# The Endemic Vascular Flora of Sardinia: A Dynamic Checklist with an Overview of Biogeography and Conservation Status

**DOI:** 10.3390/plants11050601

**Published:** 2022-02-23

**Authors:** Mauro Fois, Emmanuele Farris, Giacomo Calvia, Giuliano Campus, Giuseppe Fenu, Marco Porceddu, Gianluigi Bacchetta

**Affiliations:** 1Centre for the Conservation of Biodiversity (CCB), Department of Life and Environmental Sciences, University of Cagliari, Viale S. Ignazio da Laconi 11-13, 09123 Cagliari, Italy; mfois@unica.it (M.F.); giacomo.calvia@gmail.com (G.C.); gfenu@unica.it (G.F.); porceddu.marco@unica.it (M.P.); bacchet@unica.it (G.B.); 2Department of Chemistry and Farmacy, University of Sassari, Via Piandanna 4, 07100 Sassari, Italy; 3Independent Researcher, Via G. Rossini 69, 09045 Quartu Sant’Elena, Italy; 54.campus@libero.it; 4Sardinian Germplasm Bank (BG-SAR), Hortus Botanicus Karalitanus (HBK), University of Cagliari, Viale S. Ignazio da Laconi, 9-11, 09123 Cagliari, Italy

**Keywords:** Mediterranean flora, endemic plants, IUCN assessments, island biogeography, plant conservation, Tyrrhenian Islands, biological forms, plant evolution and distribution, plant diversity

## Abstract

The vascular flora of Sardinia has been investigated for more than 250 years, with particular attention to the endemic component due to their phylogeographic and conservation interest. However, continuous changes in the floristic composition through natural processes, anthropogenic drivers or modified taxonomical attributions require constant updating. We checked all available literature, web sources, field, and unpublished data from the authors and acknowledged external experts to compile an updated checklist of vascular plants endemic to Sardinia. Life and chorological forms as well as the conservation status of the updated taxa list were reported. Sardinia hosts 341 taxa (15% of the total native flora) endemic to the Tyrrhenian Islands and other limited continental territories; 195 of these (8% of the total native flora) are exclusive to Sardinia. Asteraceae (50 taxa) and Plumbaginaceae (42 taxa) are the most representative families, while the most frequent life forms are hemicryptophytes (118 taxa) and chamaephytes (106 taxa). The global conservation status, available for 201 taxa, indicates that most endemics are under the ‘Critically Endangered’ (25 taxa), ‘Endangered’ (31 taxa), or ‘Least Concern’ (90 taxa) IUCN categories. This research provides an updated basis for future biosystematics, taxonomic, biogeographical, and ecological studies and in supporting more integrated and efficient policy tools.

## 1. Introduction

The concept of ‘endemism’ is apparently simple but is actually problematic [1]. Previous studies on Mediterranean endemic vascular plants have led to two main approaches in defining this concept. The first approach attempted to give weight to the time of origin of a given endemic taxon [2,3]. On this basis, palaeoendemic (systematically isolated species), schizoendemic (phenotypically and karyologically similar species originated by allopatric speciation from a widespread ancestor), and neoendemic (recently diversified species or still diversifying lineages) taxa were defined. However, there is lack of phylogenetic and phylogeographic studies for many taxa, and thus the feasibility of this approach is far from being achieved, despite the increasing number of recent scientific papers that have shed light on this crucial aspect (e.g., [4,5,6,7,8]). The first attempt to organically describe the endemic vascular flora of Sardinia (consisting in the presentation of 202 taxa on the Bulletin of the Sardinian Society of Natural Sciences by Arrigoni et al. [9]) was based on this first approach. Although a valid milestone for the taxonomic, nomenclatural, ecological, and geographical description of a huge contingent of endemic taxa, the work of Arrigoni et al. [9] did not produce a satisfactory and coherent synthesis [10], the main reason being that it was based on some merely speculative assumptions.

More recently, a second approach has prevailed, based on the current distribution range of a given taxon, regardless of its age [11,12]. This concept is often used to define taxa whose geographic range is markedly more restricted than the average distribution of systematically comparable taxa. The definition is therefore inherently relative because the scale at which a taxon is restricted has to be defined subjectively (e.g., [2,12,13,14]). According to this approach, Thompson [12] recognised four categories: narrow, disjunct, regional, and Mediterranean endemic plants.

The categories originated by the two approaches are neither comparable nor overlapping. For instance, a narrow endemic plant could be either a palaeoendemic (e.g., *Ribes sardoum* Martelli), a schizoendemic (e.g., *Silene ichnusae* Brullo, De Marco & De Marco f.), or a neoendemic (e.g., *Anchusa sardoa* (Illario) Selvi & Bigazzi) [15,16,17]. In summary, the concept of endemism more used at present (and adopted in this paper) is a function of the spatial scale used to describe the restriction of the distribution of certain taxa to a definite sized area [18]. To which territory a taxon is endemic may be defined according to the specific study aim. For instance, a checklist of plants endemic to a country or a region can be useful for policy (e.g., [19,20]), while for pure-scientific purposes, an endemic flora is usually defined on a biogeographical basis [11,21]. In both cases, richness of endemic taxa in a region is one of the fundamental features of biodiversity and is valuable in the context of nature conservation policies [22]. Assessing the conservation status and the need for management actions of endemic plants is therefore of utmost importance while allocating limited financial resources [11,19,23].

Tyrrhenian continental islands in the central-western Mediterranean (Corsica, Sardinia, Balearic, Tuscan Archipelago, and Sicily) constitute one macro-hotspot of plant diversity within the Mediterranean mega-hotspot [23]. They are recognised as glacial refugia [24] and are both a reservoir of genetic diversity for widespread Mediterranean plants [25,26] and a centre of conservation and differentiation of endemism [8,27]. Sardinia is considered a meso-hotspot within the Tyrrhenian Islands macro-hotspot of biodiversity [23], with rates of endemism ranging between 13% and 15% of the local flora [12,23,28]. For its palaeogeography, current high geological and geomorphological diversity, and degree of isolation, the island supports several ancient or relict (i.e., palaeoendemics) and recently differentiated endemic plants (i.e., neoendemics) [19,29,30]. The reported numbers of the vascular plants endemic to Sardinia vary according to the reference spatial extent and recent discoveries or taxonomic reviews. Considering only data reported during the current century, 347 taxa (including hybrids and varieties) were included in the checklist by Bacchetta et al. [31] as endemics to Sardinia and shared with the Tyrrhenian Islands. The exclusive component of the island was assessed by Bacchetta et al. [19] at 168 taxa, 139 of which were species, 23 subspecies, four varieties, and two hybrids. These numbers slightly increased to 171 exclusive taxa according to more recent updates in Cañadas et al. [23] and Fenu et al. [29]. With a different approach, related to the political boundaries, Peruzzi et al. [20] recorded 180 ‘strict regional endemic’ and 322 ‘Italian endemic’ vascular plants for Sardinia.

Sardinia hosts around 25% of the 932 taxa native to Italy that were assigned to a risk category, mostly due to human disturbance [32,33]; of these taxa, 99 are endemic to the island. However, other studies accounted for further specific demographic, ecological, and environmental causes of extinction risks such as in the case of the endemic taxa belonging to the genera *Anchusa* [16,34], *Astragalus* [35], and *Centaurea* [36,37] or in the case of specific environments such as wet habitats [38], pastures [39], coastal sand dunes [40,41], and small satellite islets [42,43]. Of the 21 vascular plants currently known in Sardinia and listed in Annex II of the Habitats Directive (43/92/EEC), 17 are endemic [44]. Furthermore, five exclusive endemics (*Aquilegia barbaricina* Arrigoni & E.Nardi, *A. nuragica* Arrigoni & E.Nardi, *Lamyropsis microcephala* (Moris) Dittrich & Greuter, *Polygala sinisica* Arrigoni, and *Ribes sardoum*) were included by the IUCN/SSC—Mediterranean Plant Specialist Group in the ‘Top 50 Mediterranean Island Plants’ to be urgently conserved [45].

In this study, we present a checklist of vascular plant taxa endemic to Sardinia. Starting from the literature and unpublished knowledge by several field surveys and suggestions of the authors and external experts, a dynamic checklist was critically defined and discussed. The main aims of this study were: (1) to present and discuss an updated critical checklist of the vascular plants endemic to Sardinia; (2) to interpret their biogeography and distribution patterns; (3) to summarize their currently known conservation status; and (4) to highlight and discuss gaps and future directions for their knowledge and conservation improvement.

## 2. Results

The checklist of vascular plants endemic to Sardinia includes 341 taxa, of which 270 are *includenda* and 71 *inquirenda* (see ‘Criteria for the selection of endemic taxa’ in Section 4); a further 16 taxa have been excluded (i.e., taxa *excludenda*). The 341 endemic taxa, representing around 15% of the native Sardinian vascular flora, are referred to as 53 families, representing 42% of the 125 families of the native Sardinian flora. There are some similarities between the dominant endemic and native families in Sardinia: in both cases, the most represented family is Asteraceae, with 248 native and 49 endemic taxa. Plumbaginaceae is one of the most important exceptions, being the second largest family in endemic taxa, with 75% of native taxa being endemic. On the other hand, Poaceae, which is the third most representative family in the total native flora (211 taxa), is only the ninth family in endemics, with just 12 taxa, equal to 6% of total native Poaceae of Sardinia (Table 1).

The life form analysis highlighted a clear predominance of hemicryptophytes (118 taxa) and chamaephytes (106), followed by geophytes (56), phanerophytes (29 nano-phanerophytes + six phanerophytes), and therophytes (26). The majority of taxa are endemic exclusively to Sardinia (195 taxa); a further 81 taxa are endemic to Sardinia and Corsica, together accounting for 81% of the total endemic vascular flora (Figure 1).

Our dataset included the global assessment of the extinction risk for 201 taxa, 81% of which are exclusive to Sardinia. The remaining 140 endemic taxa were assessed as ‘Data Deficient’ (DD; 14 taxa) or ‘Not (yet) Evaluated’ (NE; 126 taxa); 59% of *inquirenda* fell under these two last categories. A total of 90 taxa (equal to 26% of the total vascular endemic flora) were of minor or ‘Least Concern’ (LC). The downside is that 21% of the 341 endemics were Vulnerable (VU, 16 taxa), Endangered (EN, 31 taxa), or Critically Endangered (CR, 25 taxa). Among the most common families, Plumbaginaceae were generally less threatened than average (64% of them are LC), while worthy of special attention are likely the Boraginaceae (38% of them under EN or CR categories), Fabaceae (29% of them under EN or CR categories), and Ranunculaceae, with 23% of endemic taxa CR and 50% not yet evaluated (NE). Apiaceae (66%), Lamiaceae (62%), and Orchidaceae (61%) were the families with the highest percentages of taxa under the DD and NE categories (Figure 2).

## 3. Discussion

The flora of Sardinia has been the subject of botanical studies for more than 250 years, starting from the first flora published by Allioni [46] to those of Moris [47], Barbey [48], Martelli [49,50], Terracciano [51,52,53], and up to the recent contribution of Arrigoni [54]. In contrast to whole flora, the endemic contingent received relatively recent specific attention, with a first attempt to achieve a comprehensive survey and synthesis less than 50 years ago [9]. After several proposals for a modern synthesis of the Sardinian endemic vascular flora [11,19,31], this research updates the 30-year-old list of Sardinian endemic vascular plants [9], incorporating recent findings, new taxa descriptions, and considering recent taxonomic revisions. Including all taxa shared with other Tyrrhenian Islands and limited portions of palaeogeographically affine continental territories, it represents the first synthesis of the entire Sardinian endemic component. Another relatively uncommon approach is to present a ‘dynamic’ checklist, which is conceived to be constantly under review and in progress. All the taxa listed here, but especially those included under doubtful status (i.e., *inquirenda*), are therefore taxa that will most likely be amended and adapted in the future in light of the expected scientific advances and discoveries. This approach has already been adopted for other recent floras (e.g., [55,56]) to broaden the debate and highlight the need for further research in even such relatively studied territories. For instance, as elsewhere in the world [57], most taxa, even several *includenda*, have been described on the basis of morphological characteristics and then never confirmed by molecular evidence. Many *inquirenda* are those cryptic and doubtful taxa belonging to the genera *Limonium*, *Ophrys*, and several Asteraceae, especially those from the *Cichorieae* tribe such as the ones of the genera *Hieracium* and *Taraxacum*. All these groups have already been outlined as problematic due to their phylogenetic complexity and instability (e.g., [58,59,60]). In particular, the above-mentioned *Limonium*, *Hieracium,* and *Taraxacum* as well as *Rubus* are genera in which apomictic reproduction is common (e.g., [58,61]).

### 3.1. Quantitative Features

Overall, our checklist accounts for 341 *includenda* and *inquirenda*, which represent around 15% of the Sardinian vascular native flora known to date (ca. 2300 taxa). A further 16 taxa were retained in a second list of *excludenda* due to the absence of recent evidence of their presence in Sardinia or taxonomic validity. However, we left a trace of their mention within the Sardinian endemic component in order to encourage possible future investigation and re-evaluation. Even though numerical comparisons with other islands are difficult, for the above-mentioned subjectivity in the definition of the endemic component, the high phytogeographic interest of the Sardinian flora was confirmed here, although geographical biases in species discovery and descriptions have been often found (e.g., [62]) and should always be considered when discussing quantitative features of species checklists. Our results are in accordance with those recently published by Thompson [12] and Médail [28], who reported similar percentages of endemic vascular plants in Sardinia. Furthermore, it is noteworthy that the percentages of endemic vascular plants exhibited by Sardinia were similar or even slightly higher than those of the other Mediterranean large islands (see summary presented in Médail [63]), and comparable to those shown by some tropical continental islands such as the large island of Hainan, which has a size similar to Sardinia (33,210 km^2^ [64]), or many small oceanic islands of Eastern Polynesia [65]. All of these comparative data sustain the inclusion of Sardinia and the other Tyrrhenian Islands within the “hottest” global hotspots of plant diversity, as already underlined by several authors [12,23,28,63,66,67].

### 3.2. Taxonomical Analysis

As in other island floras of the western Mediterranean, Asteraceae is the most represented family with 29 of the 49 endemic taxa (14% of the Sardinian endemic flora) exclusive to Sardinia. This family accounts for 25% of the Corsican endemic flora, 22% in that of Sicily, and 13% in the endemic flora of the Balearic Islands [68,69,70]. The other families (Plumbaginaceae, Fabaceae, and Caryophyllaceae), which are dominant in the Sardinian endemic flora, are also well represented in most of the other western Mediterranean islands. Within the Asteraceae, apart from the absence in Corsica of the well represented and still diversifying genus *Centaurea*, as already underlined by other authors [8], it is worth mentioning the fact that amongst the Cichorioideae, in Sardinia, there are some shrubby taxa (notably *Plagius flosculosus* (L.) Alavi & Heywood and *Buphthalmum inuloides* Moris), whereas woody Cichorioideae seem absent from other Mediterranean islands and are otherwise well represented in Macaronesia [71].

The preponderance of species belonging to the family Plumbaginaceae is mainly due to 38 taxa (34 of them exclusive to Sardinia) of the genus *Limonium* [58], which has a centre of diversification in the Mediterranean Basin (around 90% of European taxa are concentrated here). This genus comprises sexual and apomictic perennial herbs and sub-shrubs referring to the complex of species that evolved in the proto-Mediterranean area from some diploid species living on inner rocks or sea cliffs during the Oligocene and subsequently extensively diversified during the Messinian salinity crisis and Plio-Pleistocene sea-level fluctuations [72,73]. Regarding this family, it should be noted that *Armeria*, the second genus with endemic taxa (all of them exclusive to Sardinia), is present in all large western Mediterranean islands, but not in the Balearic Archipelago.

Regarding the third largest family of the Sardinian endemic flora, Fabaceae, which is underrepresented in the Corsican endemic flora [70], the Tyrrhenian Islands represent a speciation centre for many plant groups associated with Tertiary floras such as the *Genista ephedroides* DC. and *G. salzmannii* DC. groups [74,75]. This explains the presence of 16 *Genista* taxa exclusive to Sardinia.

The same is true for some genera of the fourth family, Caryophyllaceae, being that Sardinia is also a centre of diversification of the *Dianthus sylvestris* Wulfen group, nine taxa of which are exclusive to the island. The taxa belonging to this group mainly occur in rupestrian habitats, on rocky and sandy grasslands, garigues, and mesic meadows; several of them are affined to Sicilian endemic *Dianthus* taxa [76]. Even though the evolutionary processes within the group are still debated [77], these were also probably forced by the Messinian salinity crisis and by their characteristic inbreeding depression syndrome and unusual summer-flowering phenology [78,79]. Similar hypotheses were made for some of the rest of Caryophyllaceae such as taxa belonging to the *Silene mollissima* Pers. aggregate [80,81].

Worthy of mention is also the high endemism rate among the Boraginaceae family, especially due to the genus *Anchusa*, a moderate-sized genus with six of the seven allopatric endemic taxa—out of a total of about 30 taxa of the entire genus—exclusive to coastal or mountain habitats of Sardinia and with a probable common genetic pool of tertiary origin [5,16,82,83]. The high number of endemic taxa belonging to the family Boraginaceae is, especially if considering the genus *Anchusa*, exceptional with respect to the other western Mediterranean insular territories and comparable only to the number of endemic *Echium* in the Canarian Archipelago, with which Sardinia shares a Boraginaceae ancestor of Anatolian origin [5].

Although Sardinia does not host any endemic family, unlike many large oceanic and continental tropical islands (for example, Madagascar alone includes five endemic families and 321 endemic genera, see Callmander et al. [84]), the biogeographical originality of the Sardinian flora is underlined by the presence of three exclusive genera (*Castroviejoa*, *Morisia*, and *Nananthea*) and one subgenus (*Buglossites*, within the genus *Borago*), all shared with neighbouring Corsica: this sharing reinforces the vision of a unique Cyrno-Sardinian biogeographical province including the Tuscan Archipelago [19,29], where the subgenus *Buglossites* is also present [82]. The original character of the Sardinian endemic flora, compared to that of Corsica and the other large western Mediterranean islands and mainland territories, is emphasised by some very geographically and/or taxonomically isolated species, survived (or differentiated) only in Sardinia, such as *Lamyropsis microcephala* [85] and *Ribes sardoum* [86].

Few coastal endemics, mainly in the archipelagos of La Maddalena and Tavolara, are exclusive to the offshore satellite islands [87,88]. Most of them are instead shared with the coastal areas of the main Sardinian island, which were connected during the last Plio-Pleistocene eustatic fluctuations. However, these islands and islets, mostly uninhabited, act as modern refugia from the human and herbivory-related pressures [42].

### 3.3. Life Forms

Considering Raunkiaer’s life forms, the endemic flora is quite different from the whole Sardinian flora, with therophytes the dominant biological type in the whole flora but the less represented in the endemic (only 31% of endemic therophytes are exclusive to Sardinia). The low incidence of endemic therophytes is influenced by their very effective dispersal strategies [89]. Moreover, therophytes are less common in highly selective habitats and at higher elevations, where endemic taxa are most frequent [90]. As in the endemic flora of Corsica [70], hemicryptophytes are the dominant biological type among endemics, with 61% of them exclusive to Sardinia. The ratio between chamaephytes and hemicryptophytes is almost equal to one in Sardinia, while in Corsica, hemicryptophytes prevail considerably. This difference between Sardinian and Corsican components could be conditioned by different present and past climatic conditions. For both islands, high percentages of chamaephytes and hemicryptophytes confirm the crucial role of the ecological insularity of mountain massifs for the differentiation of the endemic flora [19,23,30]. In particular, the antiquity of Sardinian mountains and the high incidence of limestone cliffs have promoted the persistence of relict palaeoendemic taxa as well as more recent allopatric evolutionary processes that have given rise to many specialised chasmophytes (e.g., [88,91,92,93]). Chamaephytic and hemicryptophytic forms are also particularly adapted to the severe wind, salt spray, and aridity that are typical of coastal environments (e.g., [94,95,96]) around the about 1900 km of coastline, characterised by a variety of landforms such as cliffs, sandy dunes, long and pocket beaches, and by ca. 400 offshore satellite islands and small islets [42,43,97], also considered as important local centres of endemism (i.e., micro- and nano-hotspots *sensu* Fenu et al. [98], Cañadas et al. [23]).

Interestingly, Sardinia has more than 10% of phanerophytes (nano-phanerophytes included) in the endemic flora (whereas Corsica has less than 5%), with 35 taxa belonging to genera *Hypericum*, *Quercus*, *Rhamnus*, *Ribes*, *Rubus*, *Salix*, and especially *Genista* (18 endemic taxa, Bacchetta et al. [74,75]). Higher levels of endemic phanerophytes than Sardinian endemic flora are in Sicily, the Balearic Archipelago and Cyprus, while lower percentages characterise the floras of Corsica, Greece, southern Spain, and the Alps [68,70].

Spiny plants are also over-represented among Sardinian endemics with respect to the native flora. Other forms of adaptation to millennia of grazing/browsing pressures such as the geophytic life form, the facilitative mechanisms among plants, or their toxicity are also common (e.g., [99,100,101]), especially in the most represented families such as Asteraceae, Fabaceae, Lamiaceae, and Orchidaceae.

### 3.4. Chorological Types

The largest portion of the endemic flora is composed of exclusive taxa (SA; 57%), which, together with the taxa shared with Corsica (SA-CO; 24%), represent ca. 81% of all endemics. The exclusive Sardinian taxa amount to 8.4% of the total Sardinian flora. Compared with other islands, this percentage of exclusive taxa is lower than in Sicily (10%, Brullo et al. [68]), but higher than in Corsica (5.5%, Jeanmonod et al. [70]). The high percentage of SA and SA-CO taxa provides further evidence for the floristic autonomy of the Cyrno-Sardinian flora, mainly due to its geographical and ecological isolation that favours the in situ speciation of several taxa [4,66,70]. Interestingly, Sardinia has more exclusive endemic plants than Corsica (57% Sardinia vs. 43% Corsica), whereas Corsica has many more endemic species shared with the mainland (17.7% Corsica vs. 2.6% Sardinia) [70]. The latter difference suggests that, even if the two islands share a common floristic ancestry SA-CO taxa, accounting for 24% and 26% of the Sardinian and Corsican endemic flora, respectively, local differentiation of genetic lineages and isolation of narrow endemic species acted more in Sardinia than in Corsica. This might be because Corsica maintained more contact with continental land masses, is smaller in extension, and less surrounded by deep sea bottoms than Sardinia, which has therefore experienced longer and more repeated periods of biogeographical isolation.

Around 5% (16 taxa) of the Sardinian vascular endemic flora is composed of less common taxa shared with the Tuscan Archipelago, which constituted the land-bridge connecting the Italian Peninsula to Corsica and Sardinia during the Plio-Pleistocene eustatic fluctuations [102,103]. Within the Tuscan Archipelago, the island of Capraia (the closest to Corsica) is the one that mainly shares the Hercynian intrusive substrates—and calcifuge taxa linked to them –with Sardinia and Corsica [104]. These conditions are common in several Sardinian crystalline massifs such as Gennargentu [105], Sette Fratelli [106], or Limbara [107]. Taken together, these taxa represent the overwhelming majority of endemic vascular taxa and support the biogeographic scheme that characterises the independent Cyrno-Sardinian and Tuscan Archipelago province, already reported on the basis of faunistic and floristic approaches (e.g., [19,29,108,109,110,111]).

The remaining few—but not less interesting—endemics are those shared with the Balearic Archipelago and other Hercynian linked areas (i.e., continental and small islands in present-day Provence) or eastern and southern insular and continental territories including Sicily, the Calabrian arc in southern Italy, and the northern African regions of Kabylia and Kroumirie. These are mainly palaeoendemic relics of the pre-Oligocene connection [4,112,113], as seems to be the case for *Thymus herba**-**barona* Loisel. [6], *Soleirolia soleirolii* (Req.) Dandy [114] and *Arenaria balearica* L. [115], which often followed a well-documented east-to-west colonisation route (from the Irano-Turanian to the Anatolian plate and then to the Mediterranean), as in the cases of the ancestors of the *Lamyropsis* [85], *Arum*, *Biarum*, and *Helicodiceros* [4] genera. Other typical examples are explained by the Messinian model, when sea regressions opened lands suitable for plant colonisation and set connections among several territories, currently separated by the Mediterranean Sea [116,117].

### 3.5. Conservation Status of the Endemic Sardinian Flora

With regard to the conservation status of the endemic vascular flora, we can look at the glass as either half full or half empty. On one hand, an important percentage of endemics (26%) are under the IUCN Least Concern (LC) category. Some of them such as *Arum pictum* L.f. subsp. *pictum* or *Vinca difformis* Pourr. subsp. *sardoa* Stearn are neither rare nor threatened. Others such as *Armeria sulcitana* Arrigoni or *Ferula arrigonii* Bocchieri, although limited to a few territories, show intrinsic and extrinsic characteristics that allow them to be particularly resistant and/or resilient to pressures or to be adapted to live in remote habitats, often poor in competitors. On the other hand, 16% of the 341 endemics are either Endangered (EN) or Critically Endangered (CR). These taxa were found in decline, both in coastal and mountain areas, mainly due to human-induced habitat fragmentation and loss (e.g., [40,41,118]). Other reasons might be related to their intrinsic low genetic diversity, narrow ecological niches or population viability, which lead them to being less prone to adapt to any environmental stress and/or competition (e.g., [17,34,119,120,121]). Accordingly, several ex situ and in situ activities have been especially undertaken for those taxa currently assessed as the most endangered (e.g., [44,86,122,123,124]).

Several taxa have still not been assessed. The fact that most of them are *inquirenda* and/or having transnational distribution is not surprising. First, this suggests that propaedeutic taxonomical studies are needed to uncover the missing information about their conservation status. Genetic and phylogenetic studies have been underutilised for the establishment of priorities for conservation in most of the Mediterranean region [125]; delineating evolutionary entities, which must be those that will be subject to both ex situ and in situ conservation [126], is another crucial step towards better conservation of Sardinian flora. Second, further and larger collaborations among experts from different countries are necessary to define the global conservation status of several unassessed taxa. In consideration of the important level of regional responsibility [127] of the Sardinian community, due to the high rate of endemic vascular plants and the number of rare and/or threatened taxa [11], filling these important gaps is crucial to better address upcoming efforts in conservation planning and policy.

## 4. Materials and Methods

### 4.1. Study Area

Sardinia and its ca. 400 satellite minor islands, with a total surface area of 24,090 km^2^, is the second largest Mediterranean island after the other Italian island of Sicily. The Sardinian landscape is dominated by hilly lands, plateaus, and plains with several isolated groups of low mountains or massifs, the highest of which is the Gennargentu, with a maximum elevation of 1834 m a.s.l. The heterogeneity of substrata is mainly evidenced by Palaeozoic limestones, metamorphites and batholiths, a sedimentary lithostratigraphic complex related to a Mesozoic marine transgression, Tertiary marine and volcanic depositions related to the opening of the Tyrrhenian Basin, and Quaternary alluvial deposits [128].

Two macrobioclimates (Mediterranean and Temperate subMediterranean), four classes of continentality (from weak semihyperoceanic to weak subcontinental), eight thermotypic horizons (from lower thermoMediterranean to upper supratemperate) and seven ombrothermic horizons (from lower dry to lower hyperhumid) were identified for the entire region [129,130,131].

Six biogeographical sectors were defined based on the distribution of the endemic plants in relation to the local geology and geomorphology [29] and according to the highly diversified vegetation with 23 dynamic series (21 exclusive of Sardinia, two shared with Corsica) and five geoseries (two exclusive of Sardinia and one shared with Corsica) [130]. These sectors were hierarchically arranged within the Sardinian subprovince, the Cyrno-Sardinian and Tuscan Archipelago province, and the Italo-Tyrrhenian superprovince, which extends over the western coast of the Italian Peninsula from Liguria to Calabria [19,29]. The island possesses highly polymorphic genera that, favoured by isolation and environmental heterogeneity, are currently or recently playing key roles in the evolutionary processes of differentiation and the resulting gradual speciation of several endemic lineages such as the so-called neoendemics belonging to the genera *Limonium* [57], *Centaurea* [88], *Portulaca* [132], or *Aquilegia* [119], among others.

### 4.2. Origin and Evolution of the Endemic Component of the Sardinian Flora

The endemic plant composition—similar to most of the native flora—is shaped by the complex geologic and climatic history of western Mediterranean Basin, which has been synthesised in three main events (Figure 3).

The oldest is the mid-Tertiary fragmentation and rotation of the eastern Balearic Archipelago, Corsica, Sardinia, interior Calabrian arch, and Peloritan Massif of north-eastern part of Sicily from the current southern France and north-eastern Spain or the so-called “Protoligurian massif” [133]. The distribution of several Tyrrhenian endemic plants shared between these islands reflects this palaeogeographical event. Some examples are *Arenaria balearica* [115], *Arum pictum*, *Helicodiceros muscivorus* (L.f.) Engl. [4], *Soleirolia soleirolii*, and *Teucrium marum* L. [28,109], or the ancestors to the vicariant species belonging to genera such as *Cymbalaria* or *Thymus* [6,7]. The Messinian salinity crisis of the Late Miocene is another biogeographical event caused by the closure of the Mediterranean–Atlantic connection. This event induced an almost complete desiccation of the Mediterranean Sea and the formation of land bridges that connected, for the last time, the Cyrno-Sardinian system to the Apulian plate and the African continent [116,135,136]. During this period, some opportunities of dispersal-vicariance events and species radiations existed for drought-resistant plant species belonging to xerophytic and halophytic communities. Among the Sardinian taxa that currently show such disjunct or vicariant distributions are those of the *Cephalaria squamiflora* (Sieber) Greuter group and the *Erodium* subg. *Barbata* [117,137]. Furthermore, at the end of the Messinian salinity crisis, the relatively fast re-filling of the Mediterranean Basin caused the isolation of previously well connected plant populations that underwent a typical process of allopatric speciation, often originating groups of morphologically similar, geographically vicariant endemic species (schizoendemics): a well-known model is that of the *Silene mollissima* aggregate, which comprises 10 to 12 narrow endemic species [81,116] including the Sardinian *Silene ichnusae* and the Cyrno-Sardinian *Silene velutina* Pourr. *ex* Loisel. [121]. Finally, climatic and eustatic changes during the late Pliocene and Early-Middle Pleistocene also profoundly affected the biogeographical footprint of several Mediterranean lineages and plant assemblages, allowing for the possible terrestrial migration of a more competitive cool-temperate flora onto some offshore islands that served as glacial refugia, *sensu* Médail and Diadema [24]. The current distribution pattern of most of the endemic plants that are shared within the Cyrno-Sardinian and Tuscan Archipelago province such as *Borago pygmaea* (DC.) Chater & Greuter, *Carex microcarpa* Bertol. *ex* Moris, or *Pancratium illyricum* L. [5,82,138,139], was influenced by this event. Such as in the well-documented case of deciduous *Quercus* [140,141,142], island glacial refugia favoured the differentiation of genetic lineages of several plant groups. Some of them spread throughout the continents after the glacial ages, whereas others remained confined to the islands, giving rise to endemic taxa.

### 4.3. Criteria for the Selection of Endemic Taxa

The main criterion to discriminate endemic from non-endemic taxa was based on the biogeographical approach proposed by Médail and Quézel [66], which identified the high rate of plant endemicity of the Tyrrhenian Islands macro-hotspot as a result of a series of shared palaeogeographical events. That is, we considered a taxon as a Sardinian endemic when its distribution was confined to Sardinia or to Sardinia and at least another Tyrrhenian Island. Some taxa also present in limited neighbouring and palaeogeographically related continental territories were also included.

The initial checklist was obtained from the literature on the hitherto known flora of Sardinia [9,19,31,54] and neighbouring territories—mainly islands—whose endemic flora is partially shared with Sardinia (e.g., [143,144,145,146,147]) and integrated according to more recent specific publications, mainly focused on the taxonomic description, revision, and distribution of endemic taxa (e.g., [7,75,107,148]). Online databases such as Euro + Med Plantbase [149], Plants of the World Online [150] or the Italian FlorItaly [151], the Spanish ANTHOS [152] or the French Inventaire National du Patrimoine Naturel—INPN [153] were also consulted. We focused only on species and subspecies levels, which were nomenclaturally updated according to Bartolucci et al. [154], and critically reviewed by the authors and acknowledged external Mediterranean experts. Distribution data were mostly gleaned from the collections compiled in the second half of the 19th century, housed at CAG, CAT, FI, NAP, PAL, SASSA, SS, TO, and CJB herbaria. Further relevant distribution data as well as the Raunkiaer’s life form of each taxon were obtained from the above-mentioned literature and integrated by the experts’ unpublished knowledge. Chorological categories of the considered taxa were then defined according to their currently known distribution, following the scheme reported in Bacchetta and Pontecorvo [13] and updated in successive floras [98,105,155,156].

Since the taxonomy of several taxa is still debated and/or their presence in the island is not always sure or recently confirmed, taxa were included in the final checklist as *includenda* (i.e., currently taxonomically valid taxa, undoubtedly present in Sardinia), *inquirenda* (i.e., taxa of taxonomically doubtful validity and/or whose presence in Sardinia is in need of further investigations), or *excludenda* (i.e., currently taxonomically invalid taxa and/or whose presence in Sardinia was reported at only one time but not recognised by further research and/or never confirmed in the field in the last 50 years). Finally, all the global conservation statuses of the *includenda* and *inquirenda* taxa were retrieved from the available literature (e.g., [32]) or the IUCN database [157]. All of the mentioned information is reported for each taxon in the Appendix A.

## 5. Conclusions

This checklist confirms the phylogeographic and conservation interest of the endemic vascular flora of Sardinia. It is the result of a thorough survey of regional literature, web platforms, unpublished data, and expert opinions from the authors and other external acknowledged experts. The current state-of-the-art was resumed. Nonetheless, as almost all floras, this one is also a work in progress, which must be constantly and dynamically updated. For instance, endemic species not yet known to science can be discovered in remote places, and cryptic taxa can be hidden under the disguise of morphological similarity and will be detectable with molecular methods. These knowledge gaps need to be prioritised in the agendas of research and funding institution. Our dynamic approach illustrates converging, but also debatable perceptions and points of view, to enable more critical assessments and to emphasise the need of basic and advanced research on botany in Sardinia. Further efforts in conservation planning and policy might complete the optimistic aim of preserving this irreplaceable heritage for future generations. In this sense, we underline once again the lack of a regional law for the protection of the Sardinian flora (e.g., [19,158]), and the need of a more updated and inclusive international legal framework [159,160,161,162].

## Figures and Tables

**Figure 1 plants-11-00601-f001:**
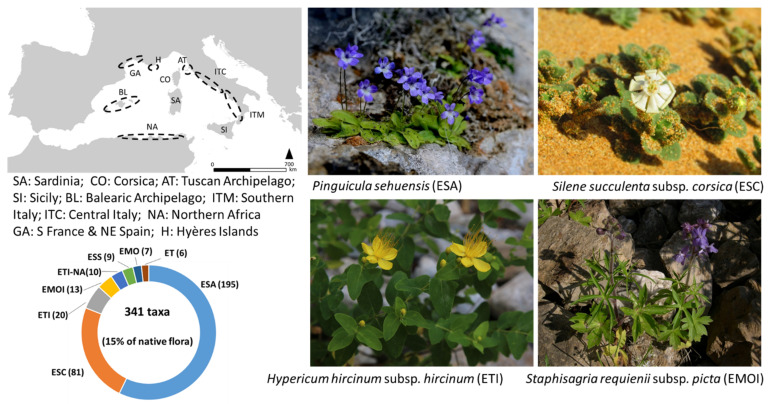
Chorological spectrum of the Sardinian endemic flora. On the right, taxa representative of the four main chorological categories: ESA (SA), ESC (SA + CO), ETI (SA ± CO + AT and/or SI), EMOI (SA ± CO ± AT + BL and/or H), ESS (SA + SI), ETI-NA (ETI + NA), ET (SA + ITM or ITC), EMO (SA + CO and/or GA, ITM, ITC).

**Figure 2 plants-11-00601-f002:**
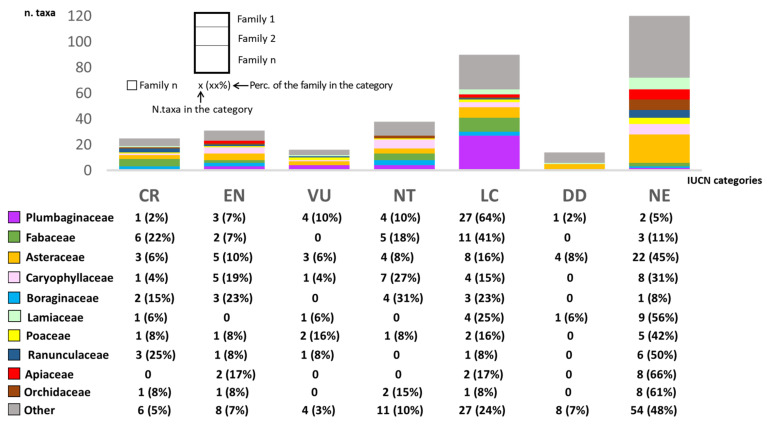
Numbers of vascular plant taxa distributed in the IUCN Red List categories by family. Only the most numerous families in each category were plotted. CR: Critically Endangered, EN: Endangered, VU: Vulnerable, NT: Near Threatened, LC: Least Concern, DD: Data Deficient, NE: Not Evaluated.

**Figure 3 plants-11-00601-f003:**
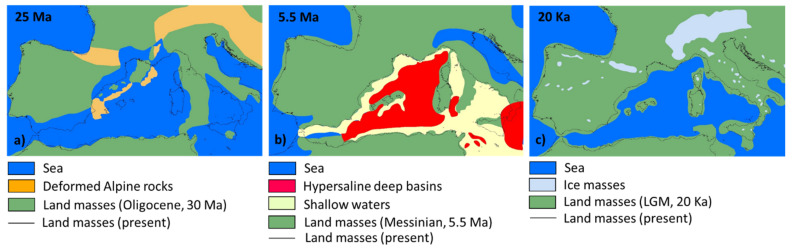
Palaeogeographical maps of the western Mediterranean Basin during the three main episodes that explain the current patterns of Sardinian plant distribution and their biogeographical links: (**a**) the backward migration of the ‘Hercynian islands’ during the Oligocene (ca. 30–15 Ma), redrawn after [4,133]; (**b**) the Messinian salinity crisis due to the closure of the Mediterranean–Atlantic gateways during the Late Miocene (ca. 6–5 Ma), redrawn after [28,134]; and (**c**) the climatic and eustatic changes of the Pleistocene, occurred from Early Pliocene (ca. 5 Ma) until the Last Glacial Maximum (LGM, ca. 20,000 years ago), redrawn after [102,103].

**Table 1 plants-11-00601-t001:** Number of endemic vascular taxa (and their relative percentage of the total taxa of the native flora) of the ten richest families in endemic species. The two most species-rich endemic genera and their respective number of doubtful taxa (i.e., *inquirenda*) are also reported for each family.

Family	No. Taxa	No. *inquirenda*	Most Representative Genera (No. Taxa)
Asteraceae	49 (20%)	19	*Hieracium* (7), *Taraxacum* (6)
Plumbaginaceae	42 (75%)	12	*Limonium* (38), *Armeria* (4)
Fabaceae	27 (12%)	0	*Genista* (18), *Astragalus* (7)
Caryophyllaceae	26 (21%)	1	*Silene* (9), *Dianthus* (9)
Lamiaceae	16 (22%)	3	*Stachys* (3), *Micromeria* (2)*, Mentha* (2)*, Teucrium* (2)
Boraginaceae	13 (28%)	2	*Anchusa* (7), *Borago* (2)
Orchidaceae	13 (20%)	4	*Ophrys* (9), *Orchis* (2)*, Serapias* (2)
Apiaceae	12 (13%)	2	*Seseli* (2), *Siler* (2)
Poaceae	12 (6%)	2	*Sesleria* (3), *Festuca* (2)*, Trisetaria* (2)
Ranunculaceae	12 (19%)	1	*Ranunculus* (5), *Aquilegia* (4)

## Data Availability

All data used in this research are available in the Appendix A.

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
