# Peer review of "The Endemic Vascular Flora of Sardinia: A Dynamic Checklist with an Overview of Biogeography and Conservation Status"

_plants, 2022, doi:10.3390/plants11050601_

Round 1
Reviewer 1 Report
Dear Editor
I have carefully read the MS which is quite well prepared for all parts. Endemism is important for the regional and global scale including a primary checklist with data analysis. In addition, supplementary materials are also well prepared. I suggested little things in the original pdf file. Finally, my suggestion is the current MS is possible to publish in the Plants journal.
Best regards,

Author Response
#Reviewer1
Dear Editor
I have carefully read the MS which is quite well prepared for all parts. Endemism is important for the regional and global scale including a primary checklist with data analysis. In addition, supplementary materials are also well prepared. I suggested little things in the original pdf file. Finally, my suggestion is the current MS is possible to publish in the Plants journal.
Best regards,
RESPONSE: Dear Reviewer, we are grateful for your positive feedback and comments. According to all suggestions of both reviewers, the manuscript has been reworked. Please, see the main text with all changes marked up using the “Track Changes” function.
Please provide the author's name following all mentioned species in whole text.
RESPONSE: as suggested, we have provided the author’s names of all species cited for the first time.
L77: why Bold?
RESPONSE: we checked the original submitted file and the text was not in bold. There might be a problem with the text visualization.

Reviewer 2 Report
The present constitutes an excellent contribution to the knowledge of the Sardinian endemic flora in a relatively broad sense, as a basis for further research in phylogeography, taxonomy and conservation biology, and for this reason it is worth publishing. Moreover, the article fits with the scope of “Plants”. Therefore, I recommend its publication in this journal.
I recommend the authors should consider some (few) observations that I indicate below.
I agree with the authors' proposal for the delimitation of endemism (in a broad sense: Sardinia, Tyrrhenian islands and other limited continental territories). This is well explained and documented, even graphically. However, perhaps some readers might have doubts in the case of plants that reach some continental territories. For example Cyclamen balearicum, which occurs in the S France, is [correctly] included in the checklist. However, doubts could exist in other species (not included in the checklist) that would have a (more or less) similar biogeographic pattern, for example Asplenium balearicum, Serapias olbia (this is a doubtful case). The above species are listed (among others) as endemic to Corsica (in a broad sense, also present in Sardinia and nearby continental territories) by Gamisans & Marzocchi (la Flore endémique de Corse). I assume that it is not easy to establish a geographical threshold in these cases. On the contrary, it is (relatively) easy to know the plants that only exist on an island, (strict endemism).
This is just a comment, I do not consider it necessary for the authors to modify their list of endemic taxa, but perhaps, in some parts of the ms, the results could be focused, with a little more detail, on the endemic plants of Sardinia in the strict sense.
Regarding these endemic taxa in the strict sense, now it is indicated that there are 195, but I have calculated 194 from the EXCEL file (I could be wrong). I recommend indicating (in the abstract) the % of Sardinia endemic taxa (in the strict sense) in the same way that is provided [15%] for the Tyrrhenian islands and other limited continental territories;
It is very correct and useful to indicate those species of taxonomically doubtful validity and/or whose presence in Sardinia is in need of further investigations (“inquirenda”). This may promote future studies that allow elucidating the taxonomic validity of some taxa or their real presence in Sardinia.
Regarding to the “inquirenda” taxa, I included here comments for some species currently listed as “includenda” in the checklist:
1) in my opinion Silene martinolii should be included in “inquirenda” (or even “excluenda”). Based on the original description and its excellent iconography, it is deduced that it is a very close taxon, perhaps conspecific, with what has been usually called S. neglecta Ten.
2) The names Santolina corsica and S. insularis (both accepted in the checklist) probably refer to the same entity. See Giacò, A. & al. 2022. Diploids and polyploids in the Santolina chamaecyparissus complex (Asteraceae) show different karyotype asymmetry, Plant Biosystems - An International Journal Dealing with all Aspects of Plant Biology, DOI: 10.1080/11263504.2022.2029971 “Preliminary morphometric, seed morpho-colorimetric, and molecular analyses highlight a strong affinity between these two taxa, suggesting that S. corsica and S. insularis could even be considered as two cytotypes of the same species (A. Giacò and collaborators, in preparation).”
I have reviewed the list of 340 accepted taxa (EXCEL file). This is an excellent, informative and updated list. I note below a few observations (based especially on my knowledge of the flora of the Balearic Islands) in case this could be useful to the authors:
Urtica atrovirens (in strict sense, not U. bianorii or U. atrovirens subsp. bianorii, endemic to Majorca) is known from Menorca; therefore perhaps it should be considered “W-Medit. subreg.” instead of “Ital.-Tyrr. superprov.”
Hyoseris taurina (Pamp.) Martinoli has been recorded from the Balearic Islands (Menorca). However this does not affect its assignment as “W-Medit. subreg.”.
See also comments on Silene martinolii and Santolina corsica, both “inquirenda” or “excludenda”
62 out of 340 taxa are “inquirenda” (I have calculated this from the EXCEL file, it should be reviewed). It is a remarkable amount. Likewise, an important proportion (20.6%) of the exclusive endemic taxa to Sardinia corresponds to “inquirenda”. Almost half of these “inquirenda” taxa (46.7%) correspond to genera in which apomictic reproduction exists or predominates: Hieracium, Taraxacum, Rubus, Limonium. If the authors consider it appropriate, perhaps some comment could be provided regarding these genera [although it is already commented very briefly only for Limonium, line 215], particularly considering that this contribution is part of a special issue “Ecology and Evolution of Plants in the Mediterranean Basin: From Knowledge to Conservation”. It is just a suggestion, the article is already very complete.
Author Response
#Reviewer 2
The present constitutes an excellent contribution to the knowledge of the Sardinian endemic flora in a relatively broad sense, as a basis for further research in phylogeography, taxonomy and conservation biology, and for this reason it is worth publishing. Moreover, the article fits with the scope of “Plants”. Therefore, I recommend its publication in this journal.
RESPONSE: Dear Reviewer, we are grateful for your positive feedback and suggestions. We reviewed the manuscript according to the Reviewers’ suggestions and after an additional careful rereading. Please, see all changes marked up using the “Track Changes” function in the main text.
I recommend the authors should consider some (few) observations that I indicate below.
I agree with the authors' proposal for the delimitation of endemism (in a broad sense: Sardinia, Tyrrhenian islands and other limited continental territories). This is well explained and documented, even graphically. However, perhaps some readers might have doubts in the case of plants that reach some continental territories. For example Cyclamen balearicum, which occurs in the S France, is [correctly] included in the checklist. However, doubts could exist in other species (not included in the checklist) that would have a (more or less) similar biogeographic pattern, for example Asplenium balearicum, Serapias olbia (this is a doubtful case). The above species are listed (among others) as endemic to Corsica (in a broad sense, also present in Sardinia and nearby continental territories) by Gamisans & Marzocchi (la Flore endémique de Corse). I assume that it is not easy to establish a geographical threshold in these cases. On the contrary, it is (relatively) easy to know the plants that only exist on an island, (strict endemism).
REPONSE: we agree with the reviewer for the inclusion of A. balearicum under the EMO chorological category. As mentioned by the Reviewer, we are not instead including S. olbia, being largely considered as invalid taxon and, even if considered as valid, we are not able to confirm its presence in Sardinia. Accordingly, the total number of endemics has been updated to 341.
This is just a comment, I do not consider it necessary for the authors to modify their list of endemic taxa, but perhaps, in some parts of the ms, the results could be focused, with a little more detail, on the endemic plants of Sardinia in the strict sense.
RESPONSE: according to the reviewer, we pointed out the percentage of endemic SA in the abstract. Moreover, we highlighted in the main text the high representativeness of this component for some genera, such as Limonium, Armeria and Anchusa or life forms, like chamaephytes and hemicryptophytes.
Regarding these endemic taxa in the strict sense, now it is indicated that there are 195, but I have calculated 194 from the EXCEL file (I could be wrong). I recommend indicating (in the abstract) the % of Sardinia endemic taxa (in the strict sense) in the same way that is provided [15%] for the Tyrrhenian islands and other limited continental territories;
RESPONSE: the reviewer is correct, as well as numbers reported in the Fig. 1. All numbers were in any case revised according to the changes suggested by the reviewer and additional small ones. Specifically, the number of SA was amended again to 195 due to the revision of L. cyrenaica, now SA.
It is very correct and useful to indicate those species of taxonomically doubtful validity and/or whose presence in Sardinia is in need of further investigations (“inquirenda”). This may promote future studies that allow elucidating the taxonomic validity of some taxa or their real presence in Sardinia.
RESPONSE: thank for the positive comment. We conceived these categories, right for the reason explained by the reviewer.
Regarding to the “inquirenda” taxa, I included here comments for some species currently listed as “includenda” in the checklist:
1) in my opinion Silene martinolii should be included in “inquirenda” (or even “excluenda”). Based on the original description and its excellent iconography, it is deduced that it is a very close taxon, perhaps conspecific, with what has been usually called S. neglecta Ten.
2) The names Santolina corsica and S. insularis (both accepted in the checklist) probably refer to the same entity. See Giacò, A. & al. 2022. Diploids and polyploids in the Santolina chamaecyparissus complex (Asteraceae) show different karyotype asymmetry, Plant Biosystems - An International Journal Dealing with all Aspects of Plant Biology, DOI: 10.1080/11263504.2022.2029971 “Preliminary morphometric, seed morpho-colorimetric, and molecular analyses highlight a strong affinity between these two taxa, suggesting that S. corsica and S. insularis could even be considered as two cytotypes of the same species (A. Giacò and collaborators, in preparation).”
RESPONSE: we agree that further studies on these taxa are needed and therefore to include them as inquirenda. As regards S. martinolii, we believe that evidences for its exclusion (i.e. as excludenda) are not yet sufficient.
I have reviewed the list of 340 accepted taxa (EXCEL file). This is an excellent, informative and updated list. I note below a few observations (based especially on my knowledge of the flora of the Balearic Islands) in case this could be useful to the authors:
Urtica atrovirens (in strict sense, not U. bianorii or U. atrovirens subsp. bianorii, endemic to Majorca) is known from Menorca; therefore perhaps it should be considered “W-Medit. subreg.” instead of “Ital.-Tyrr. superprov.”
Hyoseris taurina (Pamp.) Martinoli has been recorded from the Balearic Islands (Menorca). However this does not affect its assignment as “W-Medit. subreg.”.
See also comments on Silene martinolii and Santolina corsica, both “inquirenda” or “excludenda”
RESPONSE: we agree with all these suggestions and the checklist was accordingly amended.
62 out of 340 taxa are “inquirenda” (I have calculated this from the EXCEL file, it should be reviewed). It is a remarkable amount. Likewise, an important proportion (20.6%) of the exclusive endemic taxa to Sardinia corresponds to “inquirenda”. Almost half of these “inquirenda” taxa (46.7%) correspond to genera in which apomictic reproduction exists or predominates: Hieracium, Taraxacum, Rubus, Limonium. If the authors consider it appropriate, perhaps some comment could be provided regarding these genera [although it is already commented very briefly only for Limonium, line 215], particularly considering that this contribution is part of a special issue “Ecology and Evolution of Plants in the Mediterranean Basin: From Knowledge to Conservation”. It is just a suggestion, the article is already very complete.
RESPONSE: this comment was added in the manuscript. Thank you for the valuable suggestion